# Mixture of Experts Guided by Gaussian Splatters Matters: A new Approach to Weakly-Supervised Video Anomaly Detection

## Abstract

Video Anomaly Detection (VAD) has proved to be a challenging task due to the inherent variability of anomalous events and the scarcity of data available. Under the common Weakly-Supervised VAD (WSVAD) paradigm, only a video-level label is available during training, while the predictions are carried out at the frame-level. Despite decent progress on simple anomalous events (such as explosions), more complex real-world anomalies (such as shoplifting) remain challenging. There are two main reasons for this: (I) current state-of-the-art models do not address the diversity between anomalies during training and process diverse categories of anomalies with a shared model, thereby ignoring the category-specific key attributes; and (II) the lack of precise temporal information (*i.e.*, weak-supervision) limits the ability to learn how to capture complex abnormal attributes that can blend with normal events, effectively allowing to use only the most abnormal snippets of an anomaly. We hypothesize that these issues can be addressed by sharing the task between multiple expert models that would increase the possibility of correctly encoding the singular characteristics of different anomalies. Furthermore, multiple Gaussian kernels can guide the experts towards a more comprehensive and complete representation of anomalous events, ensuring that each expert precisely distinguishes between normal and abnormal events at the frame-level. To this end, we introduce Gaussian Splatting-guided Mixture of Experts (GS-MoE), a novel approach that leverages a set of experts trained with a temporal Gaussian splatting loss on specific classes of anomalous events and integrates their predictions via a mixture of expert models to capture complex relationships between different anomalous patterns. The introduction of temporal Gaussian splatting loss allows the model to leverage temporal consistency in weakly-labeled data, enabling more robust identification of subtle anomalies over time. The novel loss function, designed to enhance weak supervision, further improves model performance by guiding expert networks to focus on segments of data with a higher likelihood of containing anomalies. Experimental results on the UCF-Crime and XD-Violence datasets demonstrate that our framework achieves SOTA performance, scoring 91.58% AUC on UCF-Crime.

## 1 Introduction

Video Anomaly Detection (VAD) in surveillance videos is one of the most challenging tasks in the field of Computer Vision. With the increasing capabilities of deep-learning models, there have been various approaches to tackle this task. The main focus of recent research in the field of VAD has been to model spatio-temporal dependencies in videos, obtaining meaningful representations of the motion of relevant agents in the scene. In this sense, the Transformer architecture has proved to be very effective, forming the backbone of multiple works. While the current state-of-the-art models have achieved reasonable results on publicly available datasets, they still fail to capture subtle anomalies and to correctly detect the temporal window in which they happen.

We identify one of the main reasons for these issues in the formulation of the WSVAD task (Sultani et al., 2018b; Wu et al., 2022). Multi Instance Learning (MIL) strikes a balance between fully-supervised methods, which exhibit good performance but require costly data annotation, and unsu-

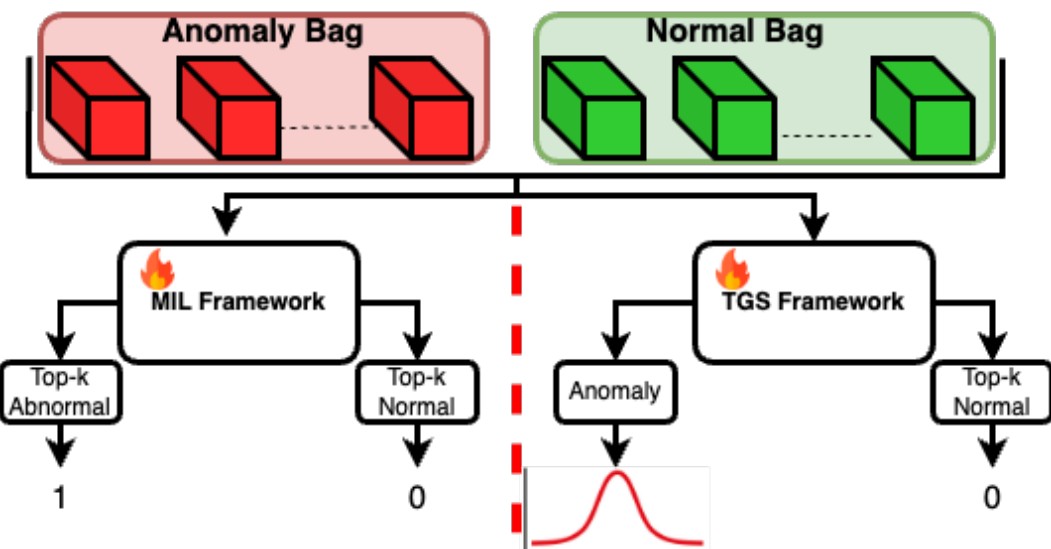

Figure 1: While SOTA methods address the task of WSVAD via the most normal and abnormal snippets in a video, the approach proposed in this paper focuses on learning a more complete representation of anomalous via Gaussian kernels.

pervised methods, which do not require manual annotations but generally result in worse performance. The core idea of MIL is to create bags containing positive and negative data samples (*i.e.*, normal and abnormal videos), labeled only at the video-level. During training, the model assigns a score between 0 and 1 to each snippet, with 0 indicating a normal snippet and 1 indicating an abnormal snippet. The highest-scoring samples in the normal bag are guided towards 0, allowing the model to learn most normal scenarios correctly. On the other hand, the highest-scoring negative samples are pushed towards 1. This leads the model to be supervised, and therefore learn, few and specific instances of anomalous events, ignoring useful information contained in neighbouring snippets and making the training process over-rely on the most abnormal snippets in a video. Furthermore, it reduces the number of more subtle anomalies on which the model is supervised. Over time, this approach has proved to be powerful but insufficient to train a model to correctly capture the secondary and specific attributes of different anomalous classes. In recent works (Yu et al., 2020; Yan et al., 2023; Georgescu et al., 2021), different auxiliary objectives are identified as priors for the VAD task in order to optimize the training process.

To address the over-reliance on the most abnormal frames, we propose to model the anomalies in a video as Gaussian distributions, rendering multiple Gaussian kernels in correspondence with peaks detected along the temporal dimension of the scores estimated for abnormal videos. This technique, called Temporal Gaussian Splatting (TSG), creates a more complete representation of an anomalous event over time, including snippets of the anomaly with lower abnormal scores in the training objective. A side-by-side comparison of the MIL task and the TGS task is shown in Figure 1. The Gaussian kernels are extracted from the abnormal scores produced by the model.

An additional challenge is related to the intrinsic differences between abnormal classes. Under the MIL paradigm, the models are trained to learn the difference between normal and abnormal videos, while the specific differences between anomalous classes are overlooked. As a result, these methods mainly focus on coarse-level representations of anomalies that allow to distinguish between normal and abnormal events, but ignore the fine-grained category-specific cues. Therefore, the more salient anomalies (*i.e.*, such as an explosion) are likely to be easily detected, while subtle anomalies (*i.e.*, shoplifting) are more likely to be confused with normal events. This constitutes a major limitation of most recent methods based on WSVAD. We address this issue via a Mixture-of-Expert (MoE) architecture, in which each expert is trained to model a single anomaly class, enhancing the specific attributes of each anomaly class that are often overlooked. To further leverage the correlations and differences between anomalies, a gate model mediates between the predictions of each expert and the more coarse-level anomalous features to learn potential interactions between anomalies.

The contributions of this paper are complementary: learning specific representations of anomalous classes allows for more accurate Gaussian kernels, and the Gaussian splatting enables the experts to

learn from more subtle anomalous events that would be overlooked otherwise. To summarize, this paper presents:

- A novel formulation of the WSVAD task based on Gaussian kernels extracted from the estimated abnormal scores to generate a more expressive and complete representation of anomalous events. Splatting the kernels along the temporal dimension allows the model to learn more precise temporal dependencies between snippets and highlight more subtle anomalies;

- A Mixture-of-Expert (MoE) architecture that focuses on individual anomaly types via dedicated class-expert models, allowing a gate model to leverage similarities and diversities between them;

- The impact of the proposed contributions is measured via an extensive set of experiments on the challenging UCF-Crime (Sultani et al., 2018a) and XD-Violence (Wu et al., 2020) datasets, showing notable improvements in performance w.r.t. previous state-of-the-art methods.

## 2 RELATED WORK

### 2.1 WEAKLY-SUPERVISED VAD

In the WSVAD task, anomalous events encompass various classes, each exhibiting distinct characteristics across the spatial and temporal dimensions. The task of WSVAD was introduced in a seminal work by (Sultani et al., 2018b). In the following years, there have been multiple different approaches that addressed the trade-off between the ease of data collection and the performance exhibited by models trained in this task. The limitation of weak labels was addressed by (Zhong et al., 2019b) using a graph convolutional network to correct noisy labels and supervise traditional anomaly classifiers. Further, (Tian et al., 2021b) proposed to learn a function of the magnitude of features to improve the classification of normal snippets and, therefore, the detection of abnormal events. The model is based on attention modules and pyramidal convolutions. The idea of improving the quality of weak labels was also explored by (Li et al., 2022c), which designed a transformer-based method trained to predict abnormal scores both at the snippet and video levels. The video-level predictions are then used to improve the performance of the model at the snippet-level. More recently, (Zhang et al., 2023b) designed a multi-head classification model that leveraged uncertainty and completeness to produce and refine its own pseudo-labels. (Majhi et al., 2024b) proposed a two-stage transformer-based model that generates anomaly-aware position embeddings and then models the short and long range relationships of anomalous events. Inspired by point-supervision (Bearman et al., 2016), (Zhang et al., 2024a) introduced Glance annotations. These annotations enhance the common weak labels by localizing a single frame in which an anomalous event is happening. While reporting very good performance, these annotations require an additional manual-labelling procedure.

Under the MIL paradigm, these variations complicate the model's ability to effectively differentiate between them. By focusing on the top-k most abnormal snippets of a video, the model is guided towards specific and evident anomalous events, without properly considering the sequence of actions that lead to them and follow them. In fact, some anomalies occur within short time windows, while others unfold over longer periods, moreover in both cases the MIL paradigm selects the same amount of abnormal snippets.

### 2.2 MIXTURE OF EXPERTS

The Mixture-of-Experts (MoE) architecture has been introduced by (Eigen et al., 2013) and has since been improved and employed for diverse tasks, from image classification to action recognition (Jain et al., 2024). The original MoE design proposed a series of small experts and a separate gate network, all receiving the same input data. Each expert predicts an output, while the gate network assigns a score of importance to them. Since then, this architecture has been improved upon by various works. A common idea across domains is to let a routing network select which portions of the input data, or input tokens, to pass to each expert (Riquelme et al., 2021; Mustafa et al., 2022; Fedus et al., 2022; Lepikhin et al., 2020). A recent work by (Puigcerver et al., 2024) proposed to weight the input tokens in a different way for each expert.

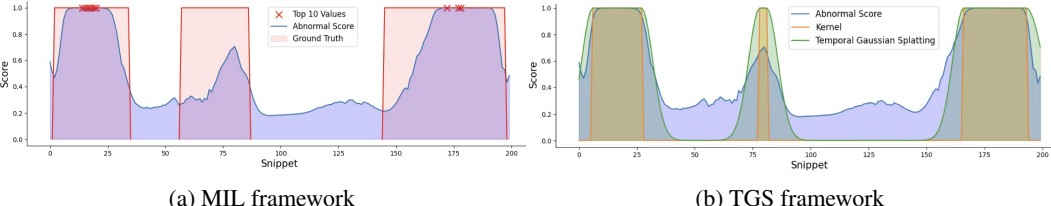

(a) MIL framework                  (b) TGS framework

Figure 2: **(a)** The abnormal scores obtained from the backbone model on a training video at the end of training. The $top_k$ snippets used in the MIL paradigm lead the model to focus on the first and last of the three anomalous events present in the video, overlooking the second anomaly. However, the second anomaly, while not scoring as high as the others, is still detected. **(b)** The Gaussian kernels extracted from the abnormal scores are splatted across the width of the detected peaks. This allows the model to learn a more complete representation of the anomalous events in the video.

### 2.3 GAUSSIAN SPLATTING

Gaussian Splatting has received a lot of attention in recent years, proving to be very effficient in fields like 3D scene reconstruction (Kerbl et al., 2023; Kopanas et al., 2021). The main idea of Gaussian Splatting is to represent each 3-dimensional point in a scene as a multivariate normal distribution, which allows to render the scene as the sum of the contributions of all the 3-dimensional areas. Gaussian splatting has since been extended to incorporate the temporal dimension in multiple domains, for example dynamic scene rendering (Li et al., 2024b;a) and medical imaging (Zhang et al., 2024b).

## 3 METHODOLOGY

Our novel Gaussian Splatter-guided Mixture-of-Experts (GS-MoE) framework aims to accurately detect complex anomalies using weakly-labeled training videos. GS-MoE leverages two key techniques: **(I) Temporal Gaussian Splatter loss**, to ensure superior separability between normal and anomalous instances under weak-supervision; **(II) Mixture-of-Experts (MoE) architecture**, that learns class-specific representations and detects complex anomalies with high confidence.

### 3.1 TEMPORAL GAUSSIAN SPLATTING (TGS)

Our Temporal Gaussian Splatting (TGS) technique provides a novel formulation of the MIL optimization paradigm by leveraging Gaussian kernels. The core idea of TGS is to reduce the overdependency on the most abnormal snippets that is often the result of the classical MIL. An example of such over-dependency is shown in Figure 2a. The $top_k$ abnormal scores are the ones that would normally be used in the loss function in the MIL paradigm:

$$top_k = \underset{|K|=k}{\arg\max} \sum_{i \in K} score_i^-, \forall i \in [1, T] \tag{1}$$

$$L = L_{topk-norm} + L_{topk-abn} \tag{2}$$

where $score_i^-$ is the score of a snippet of the abnormal video $S^-$. At the end of the training, the task encoder is able to detect two out of three anomalies contained in the video as in Figure 2a, assigning a very high abnormal score to most snippets in the first and third anomalies time window. The model is not as confident on the snippets belonging to the second anomaly, due to the fact that during training it has never been supervised specifically on them, but it assigns them an anomalous score higher than the normal snippets of the video. Additionally, the snippets between anomalies are still considered partially anomalous. We conjecture that it is possible to leverage those situations to generate pseudo-labels that allow a model to be trained on more information, while remaining in the data-annotation boundaries of the WSVAD paradigm. To do so, we employ a technique called Temporal Gaussian Splatting (TSG). Gaussian kernels are extracted in correspondence of peaks in the temporal axis of the abnormal scores predicted by a model. This allows to identify subtle

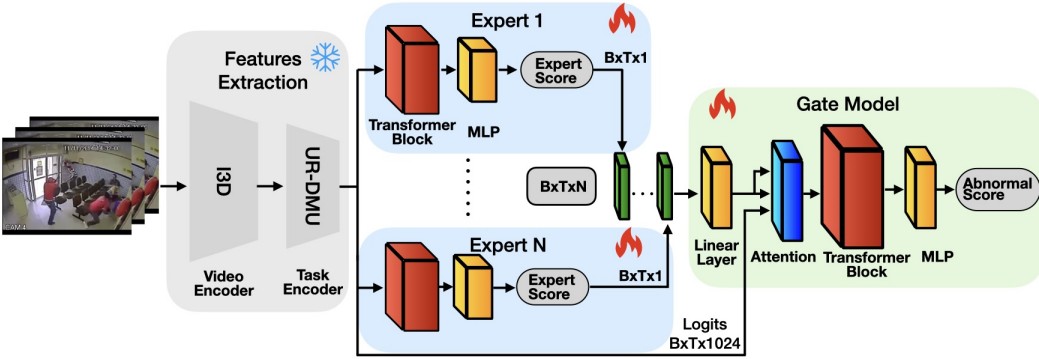

Figure 3: Overview of the GS-MoE architecture. First, in the feature extraction stage, the video encoder extracts snippet-level features from the video and the task encoder refines them in the anomaly-detection latent space. In the second stage, each expert is trained only on refined features belonging to its assigned class and to the normal class. In the final stage, the gate model collects the scores assigned by each expert and compares them with the refine features of the task encoder, producing the final abnormal score.

anomalies that are usually not included in the top-k snippets described in Equation 2. The kernels obtained from the detected peaks are then rendered over the length of anomalous videos to obtain a more accurate representation of the anomalies along the temporal dimension.

Considering the abnormal scores estimated for each snippet of a video as a signal over the duration of the video $T$, the peaks $P_1, ..., P_n$ are detected and their respective widths $W_1, ..., W_n$. The set of peaks $P$ contains the position of the snippet with the highest abnormal score for each peak in the video. This may lead to the detection of spurious peaks, meaning peaks in the abnormal scores of a video that do not belong to an anomalous event. To mitigate this, the model can be trained for a few iterations with the $L_{topk-norm}$ component of the standard MIL training objective.

Gaussian kernels $G_i$ are then initialized with unitary value for the snippets corresponding to each peak $P_i$ detected in the abnormal scores of the video. To further represent the duration of the anomaly, the kernel values corresponding to snippets that are within the width $W_i$ of the respective peak are also set to 1 if their abnormal score is higher than the difference between the peak score and the standard deviation of the normal distribution centered in the peak:

$$G_{i,t} = \begin{cases} 1, & if\ t = P_i, \\ 1, & if\ score_t \geq score_{P_i} - \sigma_i \wedge t \in W_i\ , \forall t \in [1, T] \\ 0, & otherwise \end{cases} \quad (3)$$

where $score_t$ is the abnormal score assigned to snippet $t$ and $\sigma_i$ is the standard deviation of the normal distribution centered in peak "$i$". This allows to treat each anomaly separately, which is beneficial for the WSVAD task due to the fact that different anomalies have different characteristics along the temporal dimension. Computing the Gaussian kernels in this way represent an improvement upon the top-k formulation, allowing the model to learn from the entirety of an anomalous event instead of its most abnormal snippets. Each kernel is splatted via:

$$f_i(t) = G_{i,t} \cdot \exp(-\frac{\|t - P_i\|^2}{2\sigma_i^2}), \forall t \in [1, T] \quad (4)$$

where $T$ is the length of the video and $\sigma_i$ is the standard deviation of the scores around the peak centered in $P_i$ within the width $W_i$. Finally, the pseudo-labels $\hat{y}$ are generated by rendering each of the $K$ extracted kernels over the length of the video:

$$\hat{y} = \|(\sum_{i=1}^{k} f_i(t))\| \quad (5)$$

An example of such pseudo-label (Temporal Gaussian Splatting) is shown in Figure 2b. The generated pseudo-labels contain a target abnormal score between 0 and 1 for each snippet in the video, allowing the model to learn the severity of each abnormal snippet. This represents a relevant improvement over the standard MIL training objective, where only the top-k snippets are pushed towards 1 in the training objective, as in Equation 2. Instead, the TGS loss function used to train the

experts and the MoE is formulated as:

$$L_{TGS} = L_{topk-norm} + BCE(y, \hat{y}) \tag{6}$$

### 3.2 Mixture of Experts (MoE)

Our Mixture-of-Experts (MoE) architecture is shown in Figure 3. The widely-adopted I3D model extracts the task-agnostic motion features from the videos. To further enrich the video features, the UR-DMU model (Zhou et al., 2023b) is employed as a task-aware feature extractor. The UR-DMU model is firstly trained on the WSVAD task with the standard MIL loss (Zhou et al., 2023a) and subsequently fine-tuned using the TSG loss in Equation 6. The task-aware features generated by the UR-DMU model contain enriched spatial and temporal information pertaining to anomalous events occurring within the video, compared to the more generic I3D features. However, these refined features are specialized only on distinguishing between normal and abnormal events, while overlooking the specific complexities of each anomaly class. In order to leverage these features effectively and differentiate between anomalous classes, in the second stage of the framework, multiple expert models are trained to identify the features relevant to detecting a specific type of anomaly. Consequently, the score predicted by an expert represents the likelihood that a given video corresponds to the expert's designated anomaly class. Each expert expands the boundaries of the coarse latent space learnt by the task encoder, learning to differentiate between normal videos and abnormal videos belonging to its assigned class. The experts are able to learn class-specific patterns and more subtle occurrences of anomalous events by focusing on their individual task. The architecture of an expert consists of a transformer block with 4 self-attention heads, followed by a two-layer MLP with GELU activation (Hendrycks & Gimpel, 2016), which outputs the estimated anomaly score for its respective anomaly type.

In the final stage of the framework, the scores generated by each expert are concatenated and the resulting tensor is passed to the gate model. As a first step, the gate model refines the expert's scores by projecting them into a higher-dimensional space. Then, the gate model learns the correlations between the fine-grained class specific logits of the experts and the coarse level abnormal logits of the task encoder. This is done via a bi-directional cross attention module, applied between the coarse and the fine-grained features.

The gate model learns to leverage similarities and differences between anomalous classes by processing the experts scores together with the coarse anomaly-aware features produced by the task encoder. Therefore, the gate model learns a more expressive representation of the latent space of the anomaly detection task. Finally, the abnormal scores are predicted via a transformer block followed by a four-layer MLP, similar to the architecture of the expert models.

## 4 Experiments

**Datasets.** We conduct our experiments on two widely-used Weakly-Supervised Video Anomaly Detection (WSVAD) datasets, namely UCF-Crime (Sultani et al., 2018a) and XD-Violence (Wu et al., 2020). Importantly, for both datasets, the training videos are annotated with only video-level labels, without access to frame-level annotations.

**Evaluation Metrics.** We adhere to the evaluation protocols established in prior works (Lv et al., 2023; Wu et al., 2024; Sultani et al., 2018a; Wu et al., 2020). To ensure comprehensive evaluation, we utilize multiple indicators, such as frame-level Average Precision (AP), Abnormal AP ($AP_A$) for XD-Violence and Area Under the Curve (AUC), Abnormal AUC ($AUC_A$) for UCF-Crime dataset. The AP and AUC metrics show the method robustness towards both normal and anomaly videos. However, $AP_A$ and $AUC_A$ allows to exclude normal videos where all snippets are labeled as normal and retain only the abnormal videos containing both normal and anomalous snippets. This poses a more meaningful challenge to the model's ability to accurately localize anomalies.

**Implementation Details.** The video features were obtained with the I3D model (Carreira & Zisserman, 2017) pre-trained on Kinetics-400 with sliding windows of 16 frames. The I3D implementation chosen is the ResNet50, which is proven to be one of the best-performing (Chen et al., 2021). The transformer blocks implemented in the experts and gate model do not have positional embeddings and class tokens. All models were implemented in PyTorch and trained on a single NVIDIA RTX A4500 GPU. The models were trained using the AdamW (Loshchilov & Hutter, 2017) optimizer.

| Model | Encoder | UCF-Crime | | XD-Violence | |
|---|---|---|---|---|---|
| | | AUC | AUC$_A$ | AP | AP$_A$ |
| *SoTA Methods With Multi-modal Features* | | | | | |
| MA (Zhu & Newsam, 2019) | C3D | 79.10 | 62.18 | - | - |
| HL-Net (Wu et al., 2020) | I3D | 82.44 | - | - | - |
| HSN (Majhi et al., 2024a) | I3D | 85.45 | - | - | - |
| TPWNG (Yang et al., 2024) | CLIP | 87.79 | - | 83.68 | - |
| PEMIL (Chen et al., 2024) | I3D+Text | 86.83 | - | 88.21 | - |
| VadCLIP (Wu et al., 2024) | CLIP | 88.02 | 70.23 | 84.15 | - |
| *SoTA Methods With RGB only Features* | | | | | |
| MIL (Sultani et al., 2018a) | C3D | 75.41 | 54.25 | 75.68 | 78.61 |
| | I3D | 77.42 | - | - | - |
| TCN (Zhang et al., 2019) | C3D | 78.66 | - | - | - |
| GCN (Zhong et al., 2019a) | TSN | 82.12 | 59.02 | 78.64 | - |
| MIST (Feng et al., 2021) | I3D | 82.30 | - | - | - |
| Dance-SA (Purwanto et al., 2021) | TRN | 85.00 | - | - | - |
| RTFM (Tian et al., 2021a) | I3D | 84.30 | 62.96 | 77.81 | 78.57 |
| CLAV (Cho et al., 2023) | I3D | 86.10 | - | - | - |
| UR-DMU (Zhou et al., 2023a) | I3D | 86.97 | 70.81 | 81.66 | 83.94 |
| SSRL (Li et al., 2022a) | I3D | 87.43 | - | - | - |
| MSL (Li et al., 2022b) | V-Swin | 85.30 | - | 78.28 | - |
| WSAL (Lv et al., 2021) | I3D | 85.38 | 67.38 | - | - |
| ECU (Zhang et al., 2023a) | V-Swin | 86.22 | - | - | - |
| MGFN (Chen et al., 2023) | V-Swin | 86.67 | - | - | - |
| UMIL (Lv et al., 2023) | CLIP | 86.75 | 68.68 | - | - |
| TSA (Joo et al., 2023) | CLIP | 87.58 | - | 82.17 | - |
| **GS-MoE (Ours)** | **I3D + Class Labels** | **91.58** (+3.56%) | **83.86** (+13.63%) | **82.89** | **85.74** |

Table 1: State-of-the-art comparisons on UCF-Crime and XD-Violence datasets. The best results are written in **bold**.

The batch size was set at 128, containing 64 normal and 64 abnormal videos. Under these conditions, the entire training procedure requires about three hours, while testing on the UCF-Crime test set requires 55 seconds. For training stability, during the first epoch the models are trained with the $L_{topk-norm}$ component of 6. For the same purposes, we employ the same smoothness and sparsity loss components as presented in (Sultani et al., 2018b).

## 4.1 STATE-OF-THE-ART COMPARISON

In our experiments, the proposed GS-MoE model outperforms prior state-of-the-art (SOTA) approaches across multiple metrics, as summarized in Table 1. On the challenging **UCF-Crime dataset**, GS-MoE achieves an AUC of 91.58%, surpassing the previous best model, VadCLIP (Wu et al., 2024), by 3.56%. This significant improvement illustrates the effectiveness of our model in detecting complex video anomalies in real-world datasets. Additionally, when considering the performance on the abnormal videos (AUC$_A$) only, GS-MoE achieves a score of 83.86%, which constitutes a remarkable 13.63% improvement over the second-best approach, UR-DMU (Zhou et al., 2023a), at 70.81%. This result supports one key hypothesis of our work: different types of anomalies require class-specific fine-representations for more effective detection. UR-DMU performance remains limited due to feature-magnitude based optimization which overlooks the subtle cues and enhances the sharp cues. However, the proposed TGS loss promotes both subtle and sharp cues to take part in the separability optimization. Further, the mixture-of-experts architecture is capable of capturing these class-specific representations, leading to substantial performance gains, especially on complex anomalies.

On the **XD-Violence** dataset, GS-MoE achieves an AP score of 82.89%, which is competitive with the best-performing multi-modal VadCLIP (Wu et al., 2024) model (84.15%). Moreover, when focusing on anomalous videos only, GS-MoE achieves an AP$_A$ score of 85.74%, outperforming the second-best approach, UR-DMU (Zhou et al., 2023a), which achieved an AP$_A$ score of 83.94%.

Since the AP metric considers both normal and anomaly videos for evaluation, the performance gets elevated by accurately predicting many normal videos.

As a result, methods performing well on the AP metric may still struggle in anomaly detection. The proposed method outperforms previous SOTA in the $AP_A$ metric, reinforcing its utility in real-world scenarios.

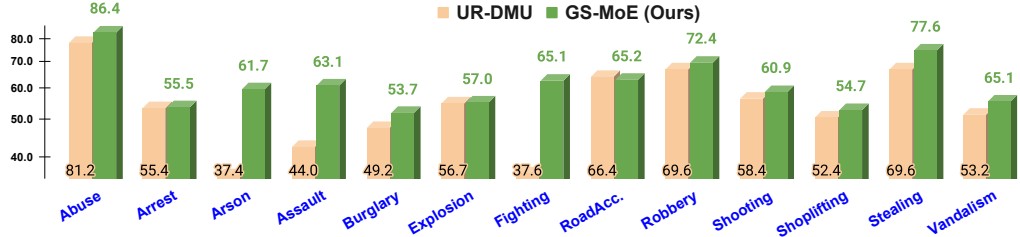

Figure 4: Category-wise performance analysis and comparison with UR-DMU.

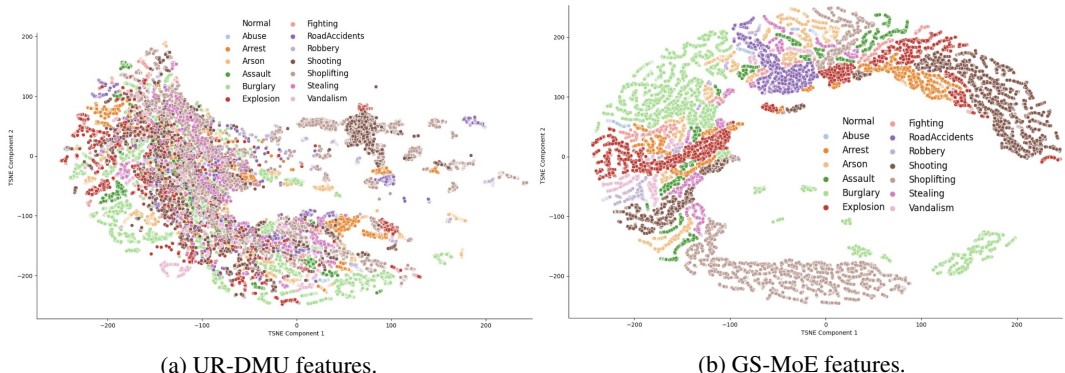

| (a) UR-DMU features. | (b) GS-MoE features. |
|---|---|

Figure 5: Category-wise t-SNE feature distribution comparison between the baseline, the experts and the gate model.

**Category-Wise Performance Analysis.** To bring additional analytical insights on the complex anomaly performance, Figure 4 provides an anomaly category-wise performance comparison between GS-MoE and the baseline UR-DMU method on the UCF-Crime dataset. Notably, significant performance boosts are recorded for complex categories like "Arson", "Assault", "Fighting", "Stealing" and "Burglary", up to +24.3%. These performance gains corroborate the benefits of GS-MoE in detecting complex video anomalies.

Figure 5 shows the t-SNE plot (van der Maaten & Hinton, 2008) of the logits obtained at each of the three stages of GS-MoE for the anomalous videos in the test set. The plot in Figure 5a, obtained from the baseline UR-DMU, shows a low degree of separability. The class diversification performed by the experts and shown in Figure 5b demonstrates the capability of GS-MoE to learn enhanced class representations.

### 4.2 QUALITATIVE RESULTS

As shown in Figure 6, the Gaussian kernels extracted from the abnormal score contain a precise representation of the anomalous events present in videos of the UCF-Crime dataset. The kernel temporal activation (heatmaps) demonstrate the capabilities of this approach. By correctly distinguishing the peaks of the anomalous events and from the spurious peaks, the model is trained to predict high anomaly scores for the associated anomalous snippets. In the "Assault-010" video sample, two peaks are detected in the abnormal score and the TGS finds a small variance for both, leading to a steep normal distribution for each of them. On the other hand, in the "Arson-011" and "Explosion-033" samples, the TGS creates much longer distributions by leading the model to estimate a large variance and producing a long time-window for the anomaly.

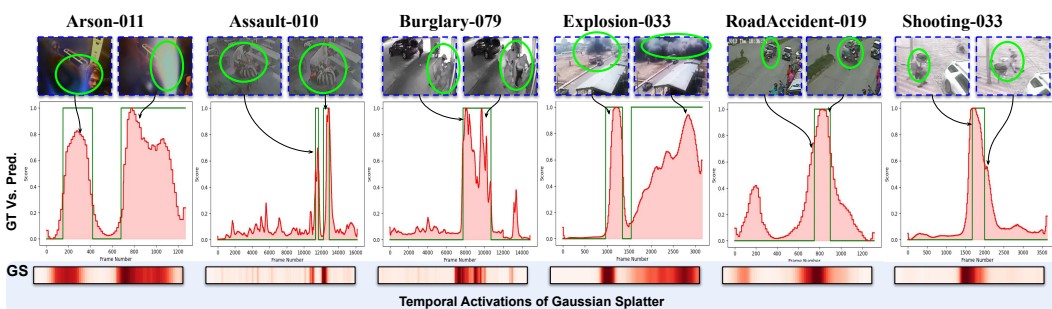

Figure 6: Visualization of sample frames and ground truth (green shed) vs. prediction scores (red shed) for various cases in Row-1 and Row-2. For each plot in Row-2, the X and Y axis denotes the number of frames and corresponding anomaly scores. Row-3 shows the temporal activation (heatmaps) learned by Gaussian splatter (GS).

## 4.3 ABLATION STUDIES

| Baseline | TSG | MoE | | AUC(%) | | $AP_A$(%) | |
|---|---|---|---|---|---|---|---|
| | | Experts | Gate | UCF-C | XD-V | UCF-C | XD-V |
| ✓ | - | - | - | 86.97 | 94.07 | 45.65 | 82.91 |
| ✓ | ✓ | - | - | 88.74 | 94.13 | 46.01 | 83.39 |
| ✓ | ✓ | ✓ | - | 89.53 | 94.29 | 47.17 | 84.16 |
| ✓ | ✓ | ✓ | ✓ | **91.58** | **94.52** | **51.63** | **85.74** |

Table 2: Impact of each component in GS-MoE framework on UCF-Crime and XD-Violence datasets.

**Component Impact:** Extensive ablation studies are conducted to evaluate the impact of each contribution to the final performance of GS-MoE, as shown in Table 2. Fine-tuning the baseline UR-DMU model with the TGS loss in Equation 6 leads to a performance increase of $+1.77\%$ on the AUC metric of UCF-Crime, while the $AP_A$ of XD-Violence increases by $+0.48\%$. These results show that the new formulation of the WSVAD task is beneficial to existing methods as well. The class-experts outperform the fine-tuned baseline by $+0.79\%$ on UCF-Crime. Notably, the $AP_A$ increases on both datasets, leading to $+1.16\%$ for UCF-Crime and $+0.76\%$ on XD-Violence, further supporting the idea that different classes of anomaly should be treated separately. Adding the gate model to the framework brings the largest performance increment. For UCF-Crime, the AUC increases by $+2.05\%$ and the $AP_A$ by $+4.46\%$. On XD-Violence, we observe relatively smaller improvements, increasing AUC by $+0.23\%$ and $AP_A$ by $+1.68\%$.

| Datasets | With task-aware features | Without task-aware features |
|---|---|---|
| **UCF-Crime** (*AUC*) | **91.58** | 90.98 |
| **XD-Violence** (*$AP_A$*) | **85.74** | 81.45 |

Table 3: Evaluation of the importance of the task-aware features for the gate model on the key metrics of the UCF-Crime and XD-Violence datasets.

**Task-Aware Features.** In order to further analyze this performance increment, the gate model was trained with and without the task-aware features. The results of this experiment are shown in Table 3. The task-aware features seem to have a key role in the performance on the $AP_A$ metric of XD-Violence. In fact, the Gate model trained with the task-aware features outperforms the other configuration by 4.29% on this setting, and by 0.6% on UCF-Crime.

**Class-Experts Impact.** The relevance of the expert models on the performance of the gate model is measured with the class-wise AUC score obtained by masking the respective class expert on the UCF-Crime dataset. The results of this experiment are shown in Table 4. By masking the

| Expert | Abuse | Arrest | Arson | Assault | Burglary | Explosion | Fighting | RoadAcc. | Robbery | Shooting | Shoplifting | Stealing | Vandalism |
|--------|-------|--------|-------|---------|----------|-----------|----------|----------|---------|----------|-------------|----------|-----------|
| Mask | 50.02 | 50.51 | 49.27 | 50.72 | 49.49 | 49.92 | 49.95 | 49.91 | 50.04 | 49.20 | 49.39 | 50.52 | 49.87 |
| W/o Mask | 86.37 | 55.48 | 61.73 | 63.12 | 53.65 | 57.04 | 65.14 | 65.22 | 72.37 | 60.89 | 54.73 | 77.62 | 57.43 |

Table 4: Category-wise performance comparison on UCF-Crime dataset between the UR-DMU baseline model and GS-MoE without the expert model for a given class. Masking the relevant experts results in an almost random output from the gate model.

experts, the measured AUC hovers around 50% for each class. On the other hand, the gate model predictions are much improved when the relevant expert score is included, leading to a significant performance boost. Most notably, the gate model scores 86.37% on the "Abuse" class, and above 70% for "Robbery" and "Stealing".

**Class experts vs cluster experts.** In practical applications, anomalies often span multiple classes, making it challenging to train a fixed set of specialized experts. To address this issue, we trained GS-MoE using cluster-based experts rather than class-specific experts. To form the data clusters, we calculated the average task-aware features for each anomalous video in the UCF-Crime training set and applied the K-Means algorithm (Lloyd, 1982) to group them. Each expert was then trained using videos from a single cluster combined with normal videos, resulting in $k$ specialized expert models. This approach enabled us to evaluate the model's performance in real-world scenarios where the number of classes is undefined. The results are reported in Table 5.

| Model | AUC |
|-------|-----|
| URDMU | 86.97 |
| TSA | 87.58 |
| TPWNG | 87.79 |
| VadCLIP | 88.02 |
| GS-MoE (5 clusters / 5 experts) | 87.35 |
| GS-MoE (6 clusters / 6 experts) | 88.03 |
| GS-MoE (7 clusters / 7 experts) | 88.58 |
| **GS-MoE (class experts)** | **91.58** |

Table 5: Comparison between the performance of GS-MoE with varying number of experts.

In this setting, GS-MoE is able to outperform current sota models by 0.56% clustering the anomalous training videos in 7 clusters and using 7 experts, while performing on par with other sota models using fewer experts. These results highlight the capabilities of GS-MoE in a real-world use-case where the number of anomalous events is not fixed.

## 5 CONCLUSION

In this work, we propose GS-MoE to provide a novel formulation for weakly-supervised video anomaly detection by leveraging Temporal Gaussian Splatting to overcome the limitations of previous methods. More specifically, we address the over-dependency on the most abnormal snippets for separability optimization. To effectively detect with the diversified categories of anomalies, our framework utilizes a mixture-of-experts architecture that learns category-specific fine-grained representations. Furthermore, it builds a correlation between the coarse abnormal cues and the learned fine-grained cues to learn a more compact representation for each category. From extensive experimentation on challenging datasets across various metrics, we find that GS-MoE consistently outperforms SOTA methods and provides new benchmark results with significant performance gains. In future works, we aim to leverage large language models to provide more explainability to abnormal categories.

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
