# SUPPLEMENTARY MATERIALS

We include additional details and results about GS-MoE. In Section 1, we report the details on the training of the models presented in the main paper as well as the data pre-processing. Section 2 presents an additional ablation study on the peak-detection mechanism, while Section 3 includes the design and experimental results on an alternative soft-moe architecture implemented for GS-MoE. Section 4 reports the computational costs of the proposed framework. In Section 5, we include a qualitative analysis of the most common failure cases of GS-MoE and finally, Section 6 contains the experimental results obtained on the UBnormal (Acsintoae et al., 2022) dataset.

## 1 IMPLEMENTATION DETAILS

The I3D features of each video have dimensions 1xNx1024, where N is the number of snippets in the video. Each snippet contains 16 consecutive frames. Each video has a different number of snippets. In order to create batches of videos, the snippets of each video are linearly projected to a fixed dimension $D$. To do so, the snippets are evenly spaced over $D$. Following the common practice in the WSVAD field, $D$ is set to 200. For example, if a video contains 100 snippets, they are projected as $[1, 1.5, 2, 2.5, ..., 99, 99.5, 100]$, where the decimal values indicate that the respective projected snippet is the weighted average between the previous snippet and the following snippet. This is done only for the training set videos, for the testing set it is not necessary to create batches. Therefore, the features dimension used as input for the expert models have dimension Bx200x1024, where B is the batch size.

**Expert Model:** The expert models are composed of a transformer block and an MLP. The input of each expert model is the 1024-dimensional logits of the task-aware encoder. The transformer block first applies layer normalization to them, followed by a 2-head self-attention layer. The output is then added to the task-aware logits and further normalized via layer normalization. The resulting tensor is projected to a 512-dimensional space and a Relu activation is applied to introduce nonlinearities in the latent space. The features obtained this way are then projected back to a 1024-dimensional space and added to the output of the self-attention layer. The MLP is composed of four linear layers than progressively reduce the dimensionality of the transformer's output to 256, 128, 64 and finally 1, which is the expert's score. A Gelu activation function is applied between the second-to-last and the last linear layer. To ensure that the score is between 0 and 1, the sigmoid function is applied to the last layer's output. An expert model contains approximately 500 thousand parameters.

**Gate Model:** The expert scores are concatenated along the last dimension, creating a tensor of dimension Bx200xN. of experts. The tensor is projected to a 1024-dimensional space via a linear layer. Then, the bi-directional cross-attention layer is applied. In one direction, it takes the projected scores as values and the task-aware logits as key and queries. In the other direction, the task-aware logits are the values and the projected score are the keys and queries. The outputs of the bi-directional attention are concatenated along the last dimension, creating a tensor of dimension Bx200x2048. This is then fed to a transformer block similar to the one described for the expert models, with 4 attention head instead of 2. This difference is due to the fact that the input of the gate's transformer block is double the dimension of the input of the expert's transformer block. The MLP component of the gate model has the same architecture as the expert's MLP. The gate model contains approximately 1 million parameters.

## 2 TGS ABLATION STUDY

As mentioned in Section 3.1, in order to mitigate the presence of spurious peaks, a model trained with TGS has to be warmed up using the standard MIL loss function or the $L_{topk-norm}$ component. In our experiments, we choose the latter. We conducted experiments with different peaks thresholds to evaluate the sensitivity of our approach to the selection of peaks. As shown in Table 1, the

performance of the model are marginally influenced by the threshold selected within the range of 0.1 and 0.3. For threshold values below 0.1, TGS detects too many peaks, especially in early stages of training, which does not allow the model to converge. On the other hand, a threshold above 0.3 leads to selecting few peaks, leading the model to estimate low scores for every video due to the fact that the major component of the loss function is given by the $top_k$ normal frames.

| | Threshold | | | | |
|---|---|---|---|---|---|
| | 0.1 | 0.15 | 0.2 | 0.25 | 0.3 |
| AUC | 90.34 | 91.08 | **91.58** | 91.23 | 90.75 |

Table 1: Performance comparison between different peak thresholds on the UCF-Crime dataset.

## 3 SOFT MOE

In order to provide an overview of the capabilities of the proposed GS-MoE framework, we implement the same training strategy with Soft-MoE, a modern MoE architecture introduced by (Puigcerver et al., 2024). The framework, shown in Figure 1, differs from the Gating model detailed in the main paper by the strategy used to leverage the expert's predictions. In the Soft-MoE architecture, the task-aware features are processed by a linear layer followed by a transformer block. A MLP predicts abnormal scores for each anomaly class in the dataset. Subsequently, these abnormal scores are weighted by the abnormal scores predicted by the experts to produce a single abnormal score.

We conducted experiments with this architecture on the UCF-Crime following the same training strategy detailed in the main paper. The results, reported in Table 2, show that the Gating model presented in the main paper achieves a 1.44% higher $AUC$ score compared to Soft-MoE. This result is in line with the results reported in Table 3 of the main paper, which highlights the benefits of processing the task-aware features together with the expert's scores.

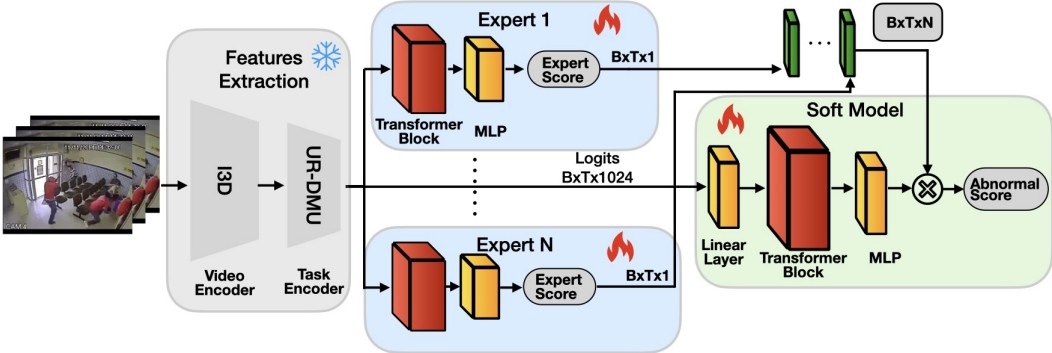

Figure 1: The Soft-MoE architecture uses the scores estimated by the experts to inform the prediction made by the gate model.

| | Gate | Soft |
|---|---|---|
| AUC | **91.58** | 90.14 |

Table 2: Comparison between the Gating MoE and the Soft MoE architectures for GS-MoE. We report the $AUC$ score achieved on the UCF-Crime dataset.

## 4 COMPUTATIONAL COSTS

The computational overhead brought by GS-MoE is reported in Table 3. GS-MoE increases the computational cost over a sota baseline model, while still able to process 10 frames per second. It is

important to notice that, in our implementation, the experts process the input in sequence. A parallel implementation would result in more frames processed per second and near real-time performance.

|  | UR-DMU | Experts | Gate | GS-MoE |
|---|---|---|---|---|
| **GFLOPs** | 1.54 | 1.56 | 0.789 | 4.133 |
| **Params. (M)** | 6.16 | 6.52 | 3.34 | 16.02 |
| **FPS** | 110.09 | 35.73 | 212.83 | 9.57 |
| *AUC* | 86.97 | 89.53 | - | **91.58** |

Table 3: Computational cost analysis for UCF-Crime with 13 experts.

## 5 QUALITATIVE RESULTS - FAILURE CASES

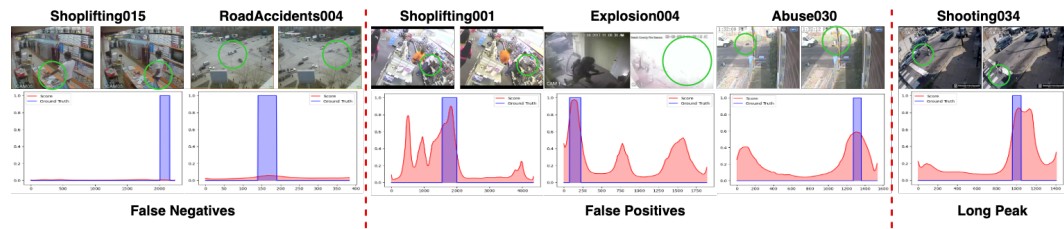

Figure 2: Failure cases examples on the UCF-Crime dataset.

In Figure 2 we report some examples of videos on which GS-MoE is unable to correctly detect the anomalous portion of the video. We identify three main failure cases: false negative, false positives and long peaks.

In WSVAD, a false negative is a missed detection of an anomaly in a video. For "Shoplifting-015" and "RoadAccidents-004", GS-MoE predicts abnormal scores close to zero for every frame. In the former example, the anomalous action is very subtle and requires a deeper understanding of the context in which the anomaly happens. Additional context cues could be useful in such cases, such as the inclusion of text features via a video-captioning model. On the other hand, in "RoadAccidents-004" the anomaly happens in a very small pixel-region of the video due to the camera being far away from the scene.

False positives are instances where GS-MoE predicts (relatively) high abnormal scores for portions of the videos that do not contain anomalous actions. The shape of the false positive peaks in the abnormal scores of "Explosion-004" and "Abuse-030" suggests that TGS could be partially responsible for them. On the other hand, in the "Shoplifting-001" video the frames in the ground-truth anomaly region closely resemble the previously ones and identifying when the anomaly starts is challenging for humans as well.

In the last example, the anomaly is correctly detected but the peak is extended further outside the anomaly region. In fact in the video of "Shooting-024", a person can be seen shooting in an empty street and then remaining on the road for a few seconds before entering a vehicle. This seems to be a common issue in videos where the anomaly action has lasting effects on the scene.

## 6 UBNORMAL EXPERIMENTS

In order to present a more comprehensive overview of the performance of GS-MoE, we experiment on the UBNormal dataset Acsintoae et al. (2022). This dataset is composed of synthetic videos generated in 29 different scenes. We experiment on this dataset in order to show the efficiency of our proposed model in data-constrained context. In fact, UBnormal contains 14.02 minutes of abnormal videos and 50.48 minutes of normal videos in the training set. The dataset does not contain anomaly-class labels, therefore we train an expert on normal and abnormal videos of a single scene,

| Model | AUC |
|---|---|
| (Georgescu et al., 2021) | 61.3 |
| (Sultani et al., 2018) | 50.3 |
| (Bertasius et al., 2021) | 68.5 |
| AnomalyRuler(Yang et al., 2025) | 71.9 |
| MULDE(Micorek et al., 2024) | **72.80** |
| GS-MoE (4 clusters / 4 experts) | 32.19 |
| GS-MoE (5 clusters / 5 experts) | 68.50 |
| GS-MoE (6 clusters / 6 experts) | 67.61 |
| **GS-MoE (7 clusters / 7 experts)** | **69.28** |
| GS-MoE (8 clusters / 8 experts) | 64.08 |
| GS-MoE (9 clusters / 9 experts) | 67.82 |
| GS-MoE (10 clusters / 10 experts) | 68.87 |
| GS-MoE (11 clusters / 11 experts) | 68.61 |
| GS-MoE (12 clusters / 12 experts) | 68.54 |
| GS-MoE (13 clusters / 13 experts) | 68.82 |
| GS-MoE (14 clusters / 14 experts) | 68.78 |
| GS-MoE (15 clusters / 15 experts) | 68.55 |
| GS-MoE (scene experts) | 65.95 |

Table 4: Performance comparison on the UBnormal dataset.

obtaining 29 scene-specialized experts. We compare the performance of our GS-MoE with other baseline models in Table 4. We also experiment by clustering the anomalous videos in the training set and assigning an expert to each cluster, as described for the UCF-Crime dataset in Section 4.3 of the main paper.

The training set of UBnormal contains 82 abnormal videos and 186 normal videos in total, but it is important to notice that there are no training abnormal videos for some scenes (scenes 7, 10 and 15), while for others there is only one anomalous video (scenes 1, 2, 5, 13, 17 and 28). This leads to a very unbalanced set of experts for the scene-experts implementation, which strongly hinders the overall performance of GS-MoE. However, GS-MoE achieves 65.95% on the $AUC$ metric in the scene-experts setting, which is better or on par with baseline methods, highlighting the efficiency of the proposed framework in such a data-constrained setting.

In the context of this dataset, the cluster-experts do not suffer from the lack of scene-specific anomalies and consistently exhibit much better performance than the scene-expert version. By clustering the training anomalous videos in seven clusters, GS-MoE is able to achieve 69.28% on the $AUC$ metric, surpassing most baseline methods albeit falling short of the sota mark.