# OpenReview forum: "Mixture of Experts Guided by Gaussian Splatters Matters: A new Approach to Weakly-Supervised Video Anomaly Detection"
_ICLR.cc/2025/Conference — Submitted to ICLR 2025_

### Official Review · Reviewer_hKhs · 2024-10-16

**Soundness:** 3
**Presentation:** 3
**Contribution:** 2
**Rating:** 3
**Confidence:** 5

**Summary:**

The authors address weakly-supervised video anomaly detection (WSVAD) by presenting a new framework termed "GS-MoE." One key aspect of their approach is mining peaks from abnormal scores and proposing "Temporal Gaussian Splatting" to generate dense pseudo-labels from these peaks for supervision. Another key point is the proposed structure of MOE, which captures the diversity of anomalies through multiple parallel Expert blocks and uses a Gate block to fuse the results of different experts. Both qualitative and quantitative results demonstrate the effectiveness of the proposed method.

**Strengths:**

1. This paper is well-written and easy to understand.
2. The application of Gaussian Splatting and MOE concepts to WSVAD is very interesting.
3. Experiments were conducted on two datasets, and the performance of the proposed method significantly surpassed that of previous methods, especially on the UCF-Crime dataset.

**Weaknesses:**

1. The core idea of Temporal Gaussian Splatting is to use Gaussian distribution to extend and smooth sparse binary snippet-level pseudo label, which has been proposed in [1].  This greatly reduces the contribution of the author's work.
2. Since the peak detection process can significantly impact the quality of the rendered anomaly score, the final performance can be greatly affected by the initialization process, including the choice of task encoder and the hyperparameters used for peak detection. Conducting more ablation studies on these factors could demonstrate the robustness of the proposed method.
3. The author mentions that the MOE architecture learns class-specific representations. However, it is known that the differentiation between experts in MOE [2] of LLM is achieved through an additional loss function. I am curious about how different experts within the proposed framework learn to focus on different anomalies using only video-level labels.
4. The ablation experiments for some important hyperparameters, such as $\sigma_i$ in Equation (3) and the number of experts $N$, were missing from the paper.
5. There are noun consistency issues in the paper, such as the use of "TGS" in Algorithm 1 and "TSG" in other parts of the paper.

[1] Zhang H, Wang X, Xu X, et al. GlanceVAD: Exploring Glance Supervision for Label-efficient Video Anomaly Detection[J]. arXiv preprint arXiv:2403.06154, 2024.

[2] Puigcerver J, Riquelme C, Mustafa B, et al. From sparse to soft mixtures of experts[J]. arXiv preprint arXiv:2308.00951, 2023.

**Questions:**

See Weakness.

---

> ### Author Response · Authors · 2024-11-22
>
> We thank the reviewer for the insightful comments and we address the weaknesses and questions raised here.
>
> Weaknesses:
>
> - GlanceVAD uses frame-level annotations to select the peaks of the gaussian distribution and then performs gaussian splatting based on fixed, per-dataset, hyperparameters (as shown in Figure 4 of the GlanceVAD paper). The frame-level annotation is advantageous and give the model additional information on the position and the quantity of anomaly events in a video. This Glance VAD setting, significantly differs from the standard WSVAD task, where there is only a single, video-level label is provided. Further, thanks to TGS, a WSVAD method can significantly push it’s limit to obtain 91.58% on challenging UCF-Crime dataset, the Glance VAD achieves 91.96% (slightly more) despite having access to labor intensive frame-level point supervision. This corroborates the contribution and effectiveness of our TGS loss towards WSVAD.
>
> - We have conducted such experiments and added a section about it in the supplementary material. By setting the prominence of peaks to values lower than 0.1, TGS would select too many peaks, making the training process more difficult. On the other hand, for values above 0.3, TGS would select too few peaks.
>
> - The expert are trained individually on specific anomaly types from the class labels included in the video-level labels in the datasets. For example, in the case of the UCF-Crime dataset, there are 13 experts, each associated to one of the 13 anomaly classes in the dataset. It is important to underline that the experts are trained individually in their warm-up phase, but when the gate model is trained it receives the abnormal scores estimated by each expert for each video. Similarly, at inference time each expert estimates an abnormal score for the video. In fact, on the XD-Violence dataset, the experts do not improve the performance over sota as much, due to the fact that a large portion of XD-Violence videos include multiple types of anomalies (as explained in the website of the dataset).
>
> - σ_i is the standard deviation of the abnormal scores estimated within the detected peak (line 263), while N is the number of anomaly classes contained in the training dataset. Therefore, they are not hyperparameters.
>
> - We have fixed the "TGS"/"TSG" inconsistencies in the upcoming revision of the paper.

---

> > ### Comment · Reviewer_hKhs · 2024-11-25
> >
> > Thanks for your great efforts on the further reply, which partly addresses my concerns.
> > However, I still have some reservations about the proposed method:
> > - The authors mentioned that "The experts are trained individually on specific anomaly types" which implies that class-level labels were used. In contrast, other methods in Table 1 only utilized video-level labels (with unknown anomaly categories). This distinction should be highlighted in the comparison.
> > - The number of experts trained is determined by the number of anomaly categories in the training set. This approach faces limitations when anomaly types are unknown or overly numerous, such as in detecting unknown anomalies or when the network parameters become excessively large.
> > - The main difference between TGS and the Temporal Gaussian Splatting introduced in GlanceVAD lies in the choice of the Gaussian kernel. More ablation studies on the Gaussian kernel should be provided to demonstrate its robustness.
> >
> > **Additional note**: The supplementary material only includes a one-page PDF. Did the author encounter an issue during the upload process, or is it a problem on my end?

---

> > > ### Author Response · Authors · 2024-11-25
> > >
> > > We apologise for the confusion, the revised supplementary material and paper are not uploaded yet, we are still running experiments requested by reviewer KARY on the UBnormal dataset. We plan to upload the revised version of both in the next few hours.
> > >
> > > - In the revised paper we will add the class-level information next to our method in Table 1.
> > >
> > > - We thank the reviewer for this observation. In case the anomaly types are unknown, it is possible to cluster task-aware embeddings via an unsupervised clustering algorithm. The classes resulting from this process can then be used to instantiate and train cluster experts without the need for the class label. We are currently running experiments in this direction and will include the results in the final version of the paper.
> > > In the revised supplementary material we include a computational cost comparison. Having many experts does not impact dramatically on the parameter count. In fact, each expert contains 500k parameters approximately, therefore a parallel implementation of their forward passes would result in near real-time fps. For the UBnormal dataset, where there are no class labels, we used an expert for each scene, resulting in 29 experts and are still able to train GS-MoE with a batch size of 128 (64 abnormal + 64 normal) on a gpu with 24GB of memory.
> > >
> > > -  The difference between TGS and GlanceVAD can be summarised in three main points:
> > >
> > >     - Peak detection vs Data annotation: TGS does not rely on frame-level annotations but only on video-level annotations. The frame-annotations are a considerable advantage for GlanceVAD. A model trained with GlanceVAD is supervised on every anomaly instance contained in the dataset, including their precise position in the video, which adds additional information that is not available in the standard WSVAD task.
> > >     - Kernel choice: GlanceVAD uses an hyper parameter (alpha in Algorithm 1 of the GlanceVAD paper) to select which scores belong to the kernel, while TGS selects the snippets belonging to the kernel by computing the standard deviation of the abnormal scores predicted by the model without any hyper parameter.
> > >     - Splatting process: while GlanceVAD uses a hyper parameter (rho_g in Figure 4a of the GlanceVAD paper) to perform the splatting, fine-tuned for each dataset, while in TGS this process is only based on the abnormal scores produced by the model.
> > >
> > > We performed ablation studies for the peak detection component of TGS, showing low sensitivity to the prominence of peaks and its robustness to the initialisation of the Gaussian kernels. We add these experiments to the (revised) supplementary material.

---

### Official Review · Reviewer_Kjt9 · 2024-10-30

**Soundness:** 3
**Presentation:** 2
**Contribution:** 2
**Rating:** 6
**Confidence:** 5

**Summary:**

This paper presents the Gaussian Splatting-guided Mixture of Experts (GS-MoE) framework, which addresses the limitations of existing models in Video Anomaly Detection (VAD) by employing multiple expert models. Each expert is trained on specific classes of anomalies, allowing the framework to effectively encode the unique characteristics of different anomalous events. The GS-MoE framework utilizes a temporal Gaussian splatting loss, which leverages temporal consistency in weakly labeled data to enhance the identification of subtle anomalies over time. By guiding the expert networks to concentrate on segments with a higher likelihood of containing anomalies, the proposed method improves performance in the weakly supervised setting.

**Strengths:**

1. The formulation of the WSVAD task using Gaussian kernels allows for a more expressive and complete representation of anomalous events.
2. The Mixture-of-Expert (MoE) architecture focuses on individual anomaly types through dedicated class-expert models. This specialization, combined with the gate model's ability to leverage similarities and diversities among different types, enhances the model's overall effectiveness and robustness in anomaly detection.

**Weaknesses:**

1. The motivation of the proposed method is unclear.
2. The innovation of the method is limited as the backbone is largely based on existing method and multi-expert, and gated model seems common.
3. The illustrations of Figure is poor, which don't covey the central idea of the proposed method.

**Questions:**

Introduction:
1. The current issues of VAD in paragraph 2 of the introduction is not well illustrated and the readers may find confusing as the definition fo the issues is not clear.
2. The motivation of introducing Gaussian splatting is not well depicted in the introduction.
3. The information of Figure 1 is extremely, which doesn't convey the central idea of this paper sufficiently. It is suggested that the authors should combine Figure1, Figure2 and Figure3 and give the readers an overview of the motivation of main idea.

Methods:
1. The innovation of the proposed method is limited as the main backbone is largely based on UR-DMU.
2. Figure 4 should add illustrations of the meaning of the icons involved.
3. Algorithm 1 seems unnecessary to presented in separate section, as plain words can presented well.

Experiments:
1. Comparisons with state-of-the-art VAD methods in 2024 should be included in Table 1.
2. The performance of XD-Violence is relative low. The authors may justify why such phenomenon happens.
3. The introducing of multi-experts inevitably increasing the computational overhead, and may further lead to issue of overfitting. The authors should conduct analysis from these aspects.

Typos:
The authors should correct their citation format using command \citet{} or \citep{}, according to the formatting instructions.

---

> ### Author Response · Authors · 2024-11-22
>
> We appreciate the reviewer for the valuable comments on the paper. Here, we address the weaknesses highlighted and questions raised.
>
> Weaknesses:
>
> - We have reworded the portion of text to better highlight the motivation behind GS-MoE.
>
> - While the backbone used is an existing model, the multi-expert and gating approach has not been fully explored in WSVAD. Furthermore, the main contribution of this paper is the re-framing of the MIL task for WSVAD, with the introduction of the Temporal Gaussian Splatting training objective.
>
> - We have updated Figure 1 and merged Figures 2 and 3 in order to give a better overlook of the contributions of TGS and the contributions of the paper.
>
> Questions:
>
> - We have modified paragraph 2 in the introduction to better explain the limitations addressed in the paper.
>
> - We have updated paragraph 3 of the introduction to make the connection between the issues of the MIL task and the motivation of TGS.
>
> - While the backbone used is an existing model, the multi-expert and gating approach has not been fully explored in WSVAD. Furthermore, the main contribution of this paper is the re-framing of the MIL task for WSVAD, with the introduction of the Temporal Gaussian Splatting training objective.
>
> - In Figure 4, the icons indicate which models are frozen (snowflake icon) or are trainable (fire icon).
>
> - In the upcoming revisions, we have removed the alghoritm from the main paper and moved it to the supplementary material.
>
> - We have added some of the most recent sota methods to Table 1.
>
> - A large portion of XD-Violence videos contain multiple types of anomalies, with one being the most prominent one (as explained on the official XD-Violence website). Therefore, the experts cannot be trained exclusively on one type of anomaly, but often see other types. This leads to less specialised experts compared to UCF-Crime.
>
> - We have added an analysis of the computational costs of GS-MoE in the supplementary material. While a certain degree of over-fitting for the experts is to be expected due to their training process, the final prediction is made by the gate model, which is trained on all anomaly types and therefore is less susceptible to over-fitting. The performance on the test set reported in Table 1, where the video-level labels are not used, seem to indicate that there is not a negative over-fitting effect.

---

### Official Review · Reviewer_dXMv · 2024-11-04

**Soundness:** 2
**Presentation:** 2
**Contribution:** 3
**Rating:** 5
**Confidence:** 4

**Summary:**

This paper proposes a GS-MoE framework for WSVAD to address the challenges of anomaly diversity and weak supervision by using multiple expert models and a temporal Gaussian splatting loss.

**Strengths:**

The paper introduces a novel framework, GS-MoE, that effectively combines expert models to address the diverse nature of anomalies in WSVAD.
The approach is well-supported by comprehensive experiments, showing strong results and achieving state-of-the-art performance.

**Weaknesses:**

The description of the Gaussian splatting loss lacks detail, making it difficult to understand the exact mechanism by which it improves temporal consistency.
There is limited explanation regarding the selection and training of individual expert models for different anomaly types, which may affect reproducibility.
There is insufficient analysis on the computational cost and efficiency of the proposed mixture of experts approach, especially for real-time applications.
More clarity is needed on the implications of using Gaussian kernels, particularly how they impact the performance and generalization of the model.

**Questions:**

What criteria are used to assign or develop expert models for specific types of anomalies?
Are there potential limitations in using this method for scenarios where anomalies are highly similar to normal events?
Could the authors elaborate on the framework's scalability and feasibility for deployment in resource-constrained environments?

---

> ### Author Response · Authors · 2024-11-22
>
> We thank the reviewer for the valuable insights provided in their review. We will address the weaknesses and questions raised here.
>
>
> Weaknesses:
> - The experts are selected based on the anomaly class of each anomaly video in the datasets. For example, in the UCF-Crime dataset there are 13 anomalous classes, therefore GS-MoE contains 13 experts, each associated with an anomaly class. During the expert’s training phase, only the expert associated to a video’s anomaly class is updated, while when the gate model is trained every expert is
> updated as well. At test time, the video is passed through each expert, providing the gate model with abnormal scores estimated by each expert
>
> - We have added an analysis of the computational costs in the upcoming revision of the supplementary material.
>
> - (We report here a similar explanation given to reviewer 33ng.) TGS generates more comprehensive pseudo-labels, which extend coverage over anomalous events and allow the model to train on segments of events that are not captured by top-k snippets. This mitigates the common over-reliance on the most abnormal frames, a limitation of the standard MIL framework. To illustrate this improvement, we have updated Figure 2. By selecting peaks with a prominence of 0.2, we strike a balance in the training process—avoiding an excess of peaks (as seen with prominence values below 0.1) while ensuring sufficient coverage (unlike prominence values above 0.5).
>
> Questions:
>
> - The expert models have the same architecture, they are associated with an anomaly class via the video-level labels contained in each dataset and are trained following the same strategy. Therefore each expert, during the expert’s warm-up phase, learns to distinguish its assigned class of anomalies from normal events.
>
> - As showed in Figure 5, in cases as ”Shoplifting”, where abnormal frames are very similar to normal frames, GS-MoE outperforms a sota model. This highlights the capability of GS-MoE to detect subtle anomalies that may be mistaken for normal events.
>
> - The proposed method is scalable and can incrementally adapt to new abnormal classes. This is accomplished by adding new experts for the corresponding abnormal classes, training each expert individually, and fine-tuning the gate model afterward. Adaptability to image resolutions depends on the feature extractor, while GS-MoE is designed for datasets where anomaly types are explicitly defined. The computational analysis we added to the supplementary material shows that the model can perform at real-time speed on a Titan RTX GPU. With constrained resources, it is likely that the model can perform at near real-time speed with a more optimised implementation than the one used for this paper. We have added a computational analysis in the upcoming revision of the supplementary material.

---

### Official Review · Reviewer_33ng · 2024-11-05

**Soundness:** 2
**Presentation:** 3
**Contribution:** 3
**Rating:** 5
**Confidence:** 4

**Summary:**

The paper introduces a novel framework named Gaussian Splatting-guided Mixture of Experts (GS-MoE) for weakly-supervised video anomaly detection (WSVAD). The approach addresses the challenges of detecting complex, real-world anomalies in videos by leveraging a set of expert models trained with a temporal Gaussian splatting loss on specific classes of anomalous events. These experts' predictions are integrated via a mixture of expert models to capture complex relationships between different anomalous patterns. The framework is designed to leverage temporal consistency in weakly-labeled data, enabling more robust identification of subtle anomalies over time. Additionally, this paper achieves state-of-the-art performance on the UCF-Crime and XDViolence datasets.


I have read the response of the authors and the comments of other reviewers. I would keep my original score.

**Strengths:**

1. The paper proposes a novel approach to WSVAD by combining Gaussian Splatting with a Mixture of Experts (MoE) architecture, which is a creative solution to address the limitations of current models in handling complex anomalies.
2. The framework demonstrates significant improvements over previous state-of-the-art methods on benchmark datasets, which is a strong indicator of the effectiveness of the proposed method.
3. The paper provides an extensive set of experiments and ablation studies that validate the effectiveness of the proposed contributions and provide insights into the impact of each component of the framework.

**Weaknesses:**

1. The paper would benefit from a more detailed analysis of the specific contributions of each architectural choice, such as the Mixture-of-Experts (MoE) design and the Temporal Gaussian Splatting (TGS) mechanism, to the overall performance. While the results are impressive, a deeper exploration of how these components enhance the model's capabilities would strengthen the paper's technical depth and provide readers with a better understanding of the innovations' impact.
2. The paper does not sufficiently address how the GS-MoE architecture scales with increasing model size or its adaptability to different image resolutions and datasets. Providing insights into the model's scalability and flexibility would be crucial for establishing its practical applicability and robustness across various settings and data.
3. The paper could benefit from a more detailed comparison with other MoE architectures to highlight the unique aspects of the proposed approach. Such a comparison would help position the GS-MoE framework within the broader landscape of anomaly detection methods and underscore its innovative features.
4. Although the paper presents numerous visualization generation results, it lacks a discussion on failure cases. Including examples where the model underperforms or fails to detect anomalies would offer a more balanced view of the framework's limitations and areas for future improvement.

**Questions:**

1. How does the method perform on other datasets with different types of anomalies, and have they considered any strategies to improve generalization?
2. What are the computational costs associated with training and deploying the GS-MoE framework, and how do they compare to other state-of-the-art methods?
3. How does the temporal Gaussian splatting loss impact the learning process, and can the authors provide more insights into how it improves the model's ability to detect anomalies?
4. Are there any immediate plans to incorporate interpretability or explainability features into the current model to understand the decisions made by the different experts?

---

> ### Author Response · Authors · 2024-11-22
>
> We thank the reviewer for the insights provided on our work and we try to address the weaknesses and questions raised here.
>
> Weaknesses:
> - Table 2 clarifies the individual contributions of the MoE and TGS mechanisms in terms of overall performance indicator. The results highlight the complementary nature of TGS and MoE, jointly enhancing the anomaly comprehension of both sharp and subtle cues by amplifying anomaly confidence levels for a more nuanced understanding. We included further ablation studies on the prominence of the detected peaks and the MoE architecture in the supplementary materials
>
> - The proposed method is scalable and can incrementally adapt to new abnormal classes. This is accomplished by adding new experts for the corresponding abnormal classes, training each expert individually, and fine-tuning the gate model afterward. Adaptability to image resolutions depends on the feature extractor, while GS-MoE is designed for datasets where anomaly types are explicitly defined. Scalability insights are provided the upcoming, revised supplementary material. Adaptability to image resolutions depends on the feature extractor, while GS-MoE is designed for datasets where anomaly types are explicitly defined.
>
> - We implemented and experimented with the Soft-MoE architecture recently proposed by Puigcerver et al. in “From sparse to soft mixtures of experts.”. We have added a section in the supplementary material explaining our implementation and reporting the results obtained with it. The performance of this architecture is 1.5% lower on the AU C metric of UCF-Crime compared to our GS-MoE architecture, therefore we decided to leave it out of the main paper.
>
> - Examples of failure cases will be included in the revised supplementary material for a balanced perspective on the model’s limitations.
>
> Questions:
>
> -We used the UCF-Crime pre-trained model for XD-Violence, but the performance do not translate well due to the fact that XD-Violence contains movie clips with cuts and different viewpoints from the CCTV viewpoints of UCF-Crime.
>
> - We report the computational costs in terms of memory, gflops and fps in the upcoming, revised supplementary materials. In general, we report that GS-MoE does not dramatically increase the computational costs over a sota model, and the fps performance could be further improved by parallelising the execution of the expert models.
>
> - TGS generates more comprehensive pseudo-labels, extending coverage over anomalous events and enabling the model to train on portions of events not captured by top-k snippets. In this way, TGS is able to mitigate the over-reliance on the most abnormal frames that often occurs in the standard MIL framework. An example of this effect is illustrated in Figure 2 (updated in the upcoming revision). By selecting peaks with a prominence of 0.2, the training process is balanced between selecting too many peaks (for prominence values lower than 0.1) and selecting too few (prominence values higher than 0.5). We included an ablation study of this in the upcoming revised supplementary material.
>
> - While GS-MoE includes interpretable components, such as class-specific experts (Table 4), further exploration of interpretability is not within the immediate scope.

---

### Official Review · Reviewer_geVr · 2024-11-06

**Soundness:** 2
**Presentation:** 2
**Contribution:** 2
**Rating:** 5
**Confidence:** 4

**Summary:**

This paper proposes a Gaussian Splatting-guided Mixture of Experts (GS-MoE), leveraging a set of experts trained with a temporal Gaussian splatting loss on specific classes of anomalous events and integrating their predictions via a mixture of expert models to capture complex
relationships between different anomalous patterns. Temporal gaussian splatting reduces the dependencies on the most abnormal snippets.

**Strengths:**

Pros:
1. Usage of Gaussian kernels extracted from the estimated abnormal scores to generate complete representation
2. Splatting the kernels along the temporal dimension for modeling anomalous temporal dependencies
3. Dedicated class-expert models focus on individual anomaly types
4. Improvements on benchmarks.

**Weaknesses:**

Cons:
1. Some inappropriate expressisons and lack in sufficient literature review
2. Each expert is trained only on refined features belonging to its assigned class and to the
normal class. Why different classes are pre-defined like this, what if different classes are
coupled? Anomalies are unexpected, it is difficult to define classes. The mixture of experts
assumes that different anomaly types can be isolated effectively. However, in real-world
applications, anomalies might not always fit neatly into predefined classes or could be ambiguous.
This assumption may restrict generalization to unseen or blended anomaly types, particularly in
more dynamic or less structured environments.
3. The model relies heavily on the quality of extracted features from pre-trained I3D models.
If the feature extraction model underperforms or isn’t well-suited to certain video types, the
overall anomaly detection may suffer. The reliance on a fixed feature extractor may limit
adaptability across datasets with different visual characteristics.
4. The combination of Gaussian splatting with a mixture of experts might make it difficult to
interpret the model’s decision-making process. Although the framework shows strong quantitative
results, it may be challenging to explain how or why certain frames are classified as anomalous,
which could be important for fields requiring clear explanations of detection results.

**Questions:**

N/A

---

> ### Author Response · Authors · 2024-11-22
>
> We appreciate the reviewer comments on the paper. Here, we answer the questions raised by the reviewer:
>
> - Each expert is associated with a specific anomaly type based on the dataset’s video-level annotations during training. For a query video during inference, all experts evaluate the video, as described in line 314 (clarified in the upcoming revision). This means that the gate model receives the score estimated by each expert based on what each expert has learned about its assigned class during individual training. In cases with coupled anomaly types, such as the XD-Violence dataset where videos often contain multiple anomaly types, the performance improvement is smaller than on UCF-Crime due to less specialized training of the experts.
>
> - We partly agree with the reviewer observation. However, we follow standard practices of weakly-supervised MIL framework where a pre-trained feature extractor (e.g. I3D) has a pivotal role. Previous WSVAD methods have mostly benefited by the backbone that has explicit mechanism to model the motion, such as I3D. As shown in Table 1, there exist many recent methods that have the same extractor as ours. Besides I3D, no feature extractor consistently produces superior results. We have trained our model with a different feature extractor, ViFiCLIP, but it did not lead to better performance due to existence of domain gap between pre-trained dataset and our task. However, with I3D, GS-MoE outperforms previous method significantly, which corroborates the effectiveness of our method towards WSVAD.
>
> - Table 4 illustrates the impact of each expert on predictions, providing valuable insights into the types of anomalous actions that tend to be misclassified. When the Mixture of Experts (MoE) correctly identifies an anomalous action (True Positive), it aligns with the detection by the corresponding expert. However, when the appropriate expert is masked, the model struggles to effectively detect anomalies associated with that expert

---

> > ### Comment · Reviewer_geVr · 2024-11-26
> > **Discussion**
> >
> > Sorry, I still have some concerns:
> > 1. In essence, the test set is independent from training set. Suppose that there is an anomaly that has a novel attribute which did not appear during training, then your set of experts determined during training cannot capture the anomaly type.
> > 2. Does the number of experts influence computational complexity ?

---

> > > ### Author Response · Authors · 2024-11-26
> > >
> > > 1. If there is a new anomaly type in the test set, the model can still detect it but it will not be as accurate as for the classes on which there is an expert available. This happens because each expert learns to distinguish a specific anomaly from normal events, therefore a new anomaly would not fall squarely in either category. In fact, in Table 5 we report the performance obtained by masking the correct expert for each anomaly type in the test set. For each class, the resulting AUC is around 50%, while if the model was completely unable to detect an anomaly those scores would be much closer to 0 because the model would consider an unseen anomaly as normal.
> > > 2. Each expert consists of approximately 500k parameters and requires 0.12 GFLOPs, therefore adding an additional expert does not increase the computational costs of GS-MoE in a dramatic way. In fact, as requested by reviewer KARY, we trained GS-MoE on the UBnormal dataset using 29 experts and were able to train the model with bach size set to 128 on a gpu with 24 gb of memory.

---

> > > > ### Comment · Reviewer_geVr · 2024-12-02
> > > > **Discussion**
> > > >
> > > > Yes, if there is a new anomaly type in the test set, the model can still detect it but it will not be as accurate as for the classes on which there is an expert available. So maybe the proposed representation cannot well generalize to unseen types.

---

### Official Review · Reviewer_tNxK · 2024-11-06

**Soundness:** 3
**Presentation:** 3
**Contribution:** 3
**Rating:** 8
**Confidence:** 3

**Summary:**

This paper introduces GS-MoE, a novel framework for weakly-supervised video anomaly detection that combines a Mixture of Experts (MoE) architecture with Temporal Gaussian Splatting (TGS). TGS uses Gaussian kernels to capture broader temporal dependencies, enhancing the model’s ability to detect subtle and complex anomalies. The MoE architecture includes specialized expert models for different anomaly types, coordinated by a gating mechanism for fine-grained detection. Experimental results on UCF-Crime and XD-Violence datasets demonstrate state-of-the-art performance.

**Strengths:**

1. The paper is well-written and clearly structured, making the ideas easy to follow.

2. The proposed GS-MoE framework is well-motivated, effectively addressing key limitations in weakly-supervised video anomaly detection. It introduces Temporal Gaussian Splatting (TGS) to reduce over-dependency on the most abnormal snippets, allowing the model to capture subtle temporal patterns across a broader range of anomaly cues. The Mixture of Experts (MoE) architecture further enhances performance by learning category-specific, fine-grained representations and connecting these with coarse anomaly cues, resulting in a more compact and accurate anomaly representation.

3. The approach achieves state-of-the-art results on UCF-Crime and XD-Violence, setting new benchmarks and demonstrating its effectiveness across various metrics.

**Weaknesses:**

1. In Table 3, there seems to be a labeling error. The column labeled "With skip connect" should likely be labeled "With task-aware features"

**Questions:**

See the weaknesses.

---

> ### Author Response · Authors · 2024-11-22
>
> We appreciate the reviewer insights. The column names in Table 3 will be corrected in the upcoming revision.

---

### Official Review · Reviewer_KARY · 2024-11-09

**Soundness:** 3
**Presentation:** 3
**Contribution:** 2
**Rating:** 6
**Confidence:** 4

**Summary:**

This paper proposes the Gaussian Splatting-guided Mixture of Experts (GS-MoE) for weakly-supervised video anomaly detection. This method enhances the capability of class-expert models to understand the anomalous events in videos by utilizing the temporal Gaussian splatter loss, and captures the complex relationships between different anomalous patterns through a mixture-of-expert architecture. The experimental results show that this method has achieved significant performance improvements on the UCF-Crime and XD-Violence datasets. This paper proposes a promising new method for weakly-supervised video anomaly detection, and its effectiveness is proved by their experiments.

**Strengths:**

Innovation: This paper proposes a new framework for weakly-supervised video anomaly detection framework that combines Gaussian splatter loss and mixture-of-expert architecture. The idea has some novelty in the field of video anomaly detection.
Performance: Experimental results show that the proposed GS-MoE achieves better performances than existing SOTA methods on both UCF Crime and XD Violence datasets with significant improvements, especially in handling complex abnormal events.
Presentation: This paper gives a detailed description for each component of GS-MoE, including the calculation of Gaussian splatter loss and the design of the mixture-of-expert architecture. This makes the proposed idea not difficult to be comprehended.
Experiments: This paper conducts extensive experiments on two widely-used datasets and compares their proposed method with other SOTA methods. Additionally, ablation studies and category performance analysis are also given to validate the effectiveness of each component.

**Weaknesses:**

My concern about this paper mainly focuses on the computational complexity due to the use of a mixture-of-expert architecture, where the authors assign an expert for each anomaly. It seems that the proposed approach may consume a significant amount of resources. I suggest the authors discuss on the inference speed of the proposed model (such as FPS) as well in Table 1, so that the advantages and disadvantages of the model can be better illustrated.

**Questions:**

1. Considering that the Gaussian splatter extracts the Gaussian kernels from the peak values of the abnormal scores and renders it onto the curve, how are the peaks detected or selected? Please give more detailed explanation about the method of peak selection, especially when the curve fluctuates frequently?
2. Considering that misclassifications may often occur in MIL, please provide some analysis of such examples, such as incorrectly detected peaks on normal snippets.
3. Please provide experimental results on the UbNormal dataset to fully demonstrate the effectiveness of the proposed method.

---

> ### Author Response · Authors · 2024-11-22
>
> The reviewer raises some valid points.
>
> - We will a table in the supplementary material to show the computational performance of GS-MoE. The AUC metric of GS-MoE shows that the trade-off between computation and performance is advantageous, with relatively low memory and GFLOPs increments. Parallelising the execution of the experts would lead to higher FPS processed.
> - We appreciate this observation and have revised the text for clarity. Peaks are identified based on a prominence threshold of 0.2. In general, a prominence lower than 0.1 leads to too many false positives, while a threshold over 0.3 leads to too few peaks detected in the early stages of training. However, we conducted further experiments in the 0.1 and 0.3 range, and showed that the training
> process is not too sensitive for prominence thresholds in this range. We include these experiments in the supplementary material. To handle frequent fluctuations, the model undergoes a warm-up phase using the standard MIL loss (topk_{normal}), which rapidly pushes obvious normal snippets toward zero. This process reduces false positives by ensuring that normal snippets close to abnormal ones, or two abnormal peaks within a single event, merge their splatted kernels over time.
> - We will include qualitative failure cases in the supplementary material for further analysis. In general, the class-wise analysis in Figure 5 shows that GS-MoE improves over SoTA on the most difficult classes of UCF-Crime such as shoplifting.
> - Experiments on the UBnormal dataset are currently being conducted and will be updated in upcoming days and also we ensure to include in the final version of the paper. However, the UBnormal dataset is not an ideal dataset for GS-MoE because it does not contain video-level anomaly type labels. We are currently experimenting with using an expert for each scene (29 scenes = 29 experts) but the
> dataset contains very few anomaly samples for each scene, rendering the MoE approach less effective than on other datasets.

---

> > ### Comment · Reviewer_KARY · 2024-11-26
> >
> > Thanks to the authors for their responses. After reading the responses, some of my concerns have been addressed. However, there seems to be no new idea in the strategy of peaks selection, which limits the novelty of the whole work. I basically recognize the contribution of this paper, and thus I decide to keep my rating of marginally accept.

---

### Author Response · Authors · 2024-11-26
**Revision Uploaded**

We thank all reviewers for the insightful comments that they provided on our work. We have uploaded a revised version of the paper and the supplementary materials. We invite the reviewers to post further comments where necessary, the discussion period has been extended for an additional week and we would appreciated additional feedbacks.

Here we report the main differences between this revised pdfs and the original version:

Main Paper:
- In Introduction, we modified paragraphs 2 and 3 as well as Figure 1 to better highlight the limitations of the MIL approach to WSVAD and the contribution of our TGS. These changes aim to address the comments made by reviewer Kjt9.
- In Section 3, we removed the algorithm (as suggested by reviewer Kjt9).
- In Section 4, we added some sota results from 2024 (as suggested by reviewer Kjt9) and specified that we use the class-labels for GS-MoE in Table 1 (as requested by reviewer hKhs).
- In Section 5, we added an ablation study on the possibility of having cluster-experts instead of class-experts (thanks to the insight offered by reviewer hKhs in their follow-up comment). We used K-Means to cluster the task-aware features into a fixed number of clusters and assign an expert to each cluster. The results on UCF-Crime outperforms most of the current sota methods but doesn't perform as well as the class-experts version, suggesting that this is a valid direction for real-world applications. In future work, we plan to expand this approach with more sophisticated clustering algorithms and additional data modalities, in order to create more semantically meaningful clusters.

Supplementary Material:
- We have added an ablation study on the performance of GS-MoE on UCF-Crime with different peak detections (as requested by several reviewers).
- We added a section detailing a different MoE implementation based on the Soft-MoE architecture (to address the comments by reviewer 33ng). In our experiments, this implementation did not perform as well as GS-MoE, which is why we left it out of the main paper. However, it does outperform the majority of sota methods on the UCF-Crime dataset.
- We added an analysis of the computational costs of GS-MoE, as requested by most reviewers. In summary, this shows that the parameter count of GS-MoE does not drastically increase over a baseline model, but our implementation of the expert's forward pass limits the fps achievable. Implementing the forward pass of each expert in parallel could make GS-MoE perform at near real-time fps.
- We added a section with some qualitative failure cases (as suggested by reviewer KARY) and an analysis of what the issues in those cases can be.
- We added a section with experimental results on the UBnormal dataset (as requested by reviewer KARY). This dataset is not ideal for the class-experts of GS-MoE because of the lack of class-labels and the limited amount of anomaly videos in the training set. The preliminary results for the clustering approach show that GS-MoE can outperform the baseline models on this dataset, but fall short of the sota results. Further refinement of the clustering approach will definitely help in this setting by providing a better differentiation between experts.

---

### Meta-Review · Area_Chair_mtxH · 2024-12-17

**Metareview:**

This paper presents the Gaussian Splatting-guided Mixture of Experts (GS-MoE) framework, which combines expert models to address the diverse nature of anomalies in WSVAD. The strength of this paper involves well motivation, good writing, and considerable improvements on UCF-Crime and XDViolence datasets. The weakness is that the proposed method can not guarantee generalization ability on unknown abnormal classes, limiting its novelty. The authors are encouraged to explore more general forms of experts, as many anomalies in real-world scenarios are unpredictable.

**Additional Comments On Reviewer Discussion:**

The concerns raised by reviewers mainly include lack of complexity analysis, insufficient ablation studies, details and precision of peak detection, generalization ability to unknown abnormal class, idea novelty, etc. The authors provided detailed explanations and promised to conduct some suggested in the revised paper. Two reviewers (hKhs, geVr KARY) admitted that some concerns were addressed, but the generalization issues to unknown abnormal class remained unresolved. This limits the novelty of this paper.

---

### Decision · Program_Chairs · 2025-01-22

Reject